# VRSD: Rethinking Similarity and Diversity for Retrieval in Large Language Models

## Abstract

Vector retrieval algorithms are essential for semantic queries within the rapidly evolving landscape of Large Language Models (LLMs). The ability to retrieve vectors that satisfy both similarity and diversity criteria substantially enhances the performance of LLMs. Although Maximal Marginal Relevance (MMR) is widely employed in retrieval scenarios requiring relevance and diversity, variations in the parameter $\lambda$ lead to fluctuations that complicate the optimization trajectory in vector spaces. This obscures the direction of improvement and highlights the lack of a robust theoretical analysis regarding similarity and diversity constraints in retrieval processes. To address these challenges, this paper introduces a novel approach that characterizes both constraints through the relationship between the sum vector and the query vector. The proximity of these vectors ensures the similarity constraint, while requiring individual vectors within the sum vector to diverge in their alignment with the query vector satisfies the diversity constraint. We first formulate a new combinatorial optimization problem, selecting $k$ vectors from a candidate set such that their sum vector maximally aligns with the query vector, and demonstrate that this problem is **NP-complete**. This result underscores the inherent difficulty of simultaneously achieving similarity and diversity in vector retrieval, thereby providing a theoretical foundation for future research. Subsequently, we present the heuristic algorithm **V**ectors **R**etrieval with **S**imilarity and **D**iversity, **VRSD**, which features a clear optimization objective and eliminates the need for preset parameters. VRSD also achieves a modest reduction in time complexity compared to MMR. Empirical validation confirms that VRSD significantly outperforms MMR across various datasets, while also demonstrating that the sum vector effectively captures both diversity and similarity simultaneously. The data and code are available at https://anonymous.4open.science/r/VRSD-CF9D.

## 1 Introduction

Vector retrieval algorithms are crucial for semantic queries and have become increasingly integral to the deployment of Large Language Models (LLMs). Effective interaction with LLMs frequently necessitates the provision of relevant or similar examples to elicit enhanced responses (Liu et al., 2022). The introduction of Retrieval Augmented Generation (RAG) has notably advanced the capabilities in knowledge-intensive tasks (Lewis et al., 2020), underscoring the growing importance of retrieval methods. Empirical evidence suggests that employing the BM25 algorithm to select examples from the training set markedly improves LLMs performance over random selection (Liu et al., 2022; Luo et al., 2023). Moreover, leveraging existing text embedding models for example retrieval often surpasses BM25, particularly in specific contexts (Reimers & Gurevych, 2019; Wang et al., 2022). And the advent of Dense Retrieval, which employs dense vectors for semantic matching in latent spaces (Chen et al., 2017; Lee et al., 2019), represents a evolution over traditional sparse retrieval methods like BM25 by utilizing the robust modeling capabilities of pre-trained language models to learn relevance functions (Devlin et al., 2019). Innovations such as the applying the dual encoder framework (Karpukhin et al., 2020) and dynamic listwise distillation (Ren et al., 2021) have further refined the effectiveness of dense retrieval techniques. Subsequent enhancements in semantic parsing and in-context learning (Pasupat et al., 2021), facilitated by feedback from LLMs (Rubin et al., 2022), have enabled more precise example selection and improved answer accuracy. Despite ongoing advancements in retrieval methods, the broadening application scope of LLMs necessitates

retrieval approaches that balance relevance with diversity—specifically, a relevance-focused diversity rather than an unrestricted diversity. Additionally, the RAG framework's ability to augment the LLMs' external data access also underscores the need for simple yet efficient algorithms that can streamline the retrieval process.

Considering the balance between similarity and diversity, the Maximal Marginal Relevance (MMR) (Carbonell & Goldstein, 1998) is an effective algorithm and has been widely applied in vector retrieval practices. Aiming to achieve an optimal balance, MMR incorporates a parameter, $\lambda$, which adjusts the weight of relevance and diversity by varying its value. Nevertheless, this method is not always effective; in different scenarios, $\lambda$ needs to take different values, which cannot be known in advance. Recent research (Rubin et al., 2022; Wang et al., 2023) has also explored using LLMs to enhance retrieval results, while also suggests considering the selection of a set of examples from a combinatorial optimization perspective, rather than selecting examples one by one, as the in-context examples can influence each other. In light of this, we propose using the sum vector to characterize both similarity and diversity in vector retrieval. Simply put, this involves maximizing the similarity between the sum vector of the selected vectors and the query vector, and maximizing the similarity of the sum vector to the query vector imposes a similarity constraint. At the same time, from a geometric perspective, the requirement for the sum vector to be similar to the query vector means that the selected vectors approach the query vector from different directions, thus imposing a diversity constraint. Additionally, the idea of considering the similarity between the sum vector and the query vector is analogous to the famous finding in word2vec (king - man + woman = queen) (Mikolov et al., 2013), as both involve obtaining complex semantic similarities through simple vector arithmetic. Therefore, using the sum vector to characterize similarity and diversity constraints not only considers similarity while reducing redundancy but also enhances the complementarity among retrieval results.

Consequently, we define a new combinatorial optimization problem: selecting several vectors from a set of candidate vectors such that the similarity between the sum vector of the selected vectors and the query vector is maximized. However, contrary to its intuitive and straightforward appearance, this is a highly challenging problem. We prove that this problem is NP-complete by reducing the subset sum problem to it, revealing theoretically that simultaneously pursuing similarity and diversity in vector retrieval is extremely difficult. This novel combinatorial optimization problem, of independent theoretical interest, establishes a solid theoretical foundation for future research. Subsequently, we present a heuristic algorithm to solve the proposed problem. This algorithm has a clear optimization objective, requires no preset parameters, and has a slightly lower time complexity than the MMR algorithm. Our experimental studies also demonstrate that the new algorithm significantly outperforms the MMR algorithm across various datasets. Additionally, given that similarity measures in vector retrieval typically include cosine similarity, inner product distance, and Euclidean distance, and considering that vectors in LLM applications are usually normalized, the results obtained using these measures in vector retrieval are consistent. Consequently, the discussion on vector similarity in this paper uses cosine similarity. In summary, our work makes the following contributions:

- We propose using the sum vector to characterize similarity and diversity constraints in vector retrieval. We formulate a novel optimization problem where we seek to select several vectors from a set of candidates such that the similarity between the sum vector of the selected vectors and the query vector is maximized.
- We demonstrate that our optimization problem is NP-complete, theoretically revealing the extreme difficulty of simultaneously pursuing similarity and diversity in vector retrieval.
- For the NP-complete combinatorial optimization problem we propose, we provide a heuristic algorithm, VRSD. We experimentally study our algorithm on several datasets, and our results show that our algorithm significantly outperforms the classic MMR algorithm.

## 2 THEORETICAL ANALYSIS OF MMR

### 2.1 LIMITATIONS OF MMR

To enhance retrieval processes by accounting for both relevance and diversity, the Maximal Marginal Relevance (MMR) algorithm was introduced (Carbonell & Goldstein, 1998). MMR addresses the balance between relevance and diversity in traditional retrieval and summarization methods by employing "marginal relevance" as an evaluation metric. This metric is defined as a linear combination

of independently measured relevance and novelty, formulated as Eq.1:

$$\text{MMR} = \arg \max_{d_i \in R \setminus S} [\lambda \cdot \text{Sim}_1(d_i, q) - (1 - \lambda) \cdot \max_{d_j \in S} \text{Sim}_2(d_i, d_j)]. \tag{1}$$

The challenge lies in selecting an appropriate $\lambda$ to achieve the desired balance between relevance and diversity, particularly in high-dimensional vector spaces where the impact of varying $\lambda$ is less predictable. This variability in $\lambda$ leads to fluctuations in retrieval results, resulting in unpredictable consequences, which can be illustrated by a simple example. Commonly, $\lambda$ is preset at a value of 0.5 in many MMR implementations, a choice that stems from the algorithm's foundational design. It is important to note that at $\lambda = 1$, the algorithm exclusively prioritizes relevance, while at $\lambda = 0$, it focuses entirely on diversity. Let us

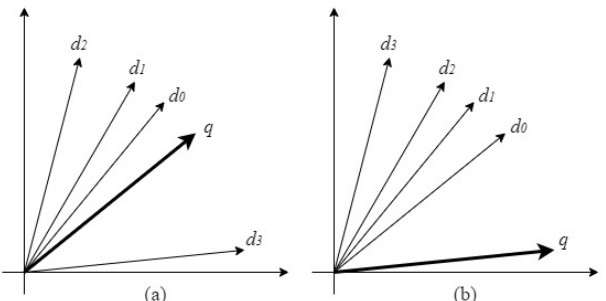

Figure 1: An analysis of the Maximal Marginal Relevance. (a) The candidate vectors are located on different sides of the query vector. (b) The candidate vectors are located on the same side of the query vector.

examines the performance of the MMR algorithm at the typical midpoint setting of $\lambda = 0.5$. For clarity and ease of comprehension, we model the retrieval process within a two-dimensional vector space, though the principles observed are equally applicable to more complex, higher-dimensional scenarios.

As illustrated in Figure.1(a), consider $q$ as the query vector, and $d_0$ to $d_3$ as candidate vectors that surpass the relevance threshold, collectively represented as $R = \{d_0, d_1, d_2, d_3\}$, with $S$ initially empty. Utilizing the MMR algorithm, $d_0$ is first selected due to its highest relevance to $q$, determined using cosine similarity as a measure. Subsequently, $d_3$ is chosen over $d_1$, despite $d_1$ having a smaller angle with $q$ and thus greater direct relevance. The selection of $d_3$ is influenced by the fact that the cumulative relevance between $d_1$ and $d_0$ significantly surpasses that between $d_3$ and $d_0$, resulting in a higher MMR value for $d_3$ as per the formula.

However, as depicted in Figure.1(b), with $q$ serving as the query vector and $R = \{d_0, d_1, d_2, d_3\}$ representing the initial set of candidate vectors, $d_0$ is first selected due to its maximal relevance to $q$. The selection process using the MMR algorithm proceeds as follows: with $\lambda = 0.5$, $S = \{d_0\}$, and $R \setminus S = \{d_1, d_2, d_3\}$, the formula can be articulated as Eq.2:

$$\text{MMR} = \arg \max_{i=1,2,3} [0.5 \cdot (\text{Sim}_1(d_i, q) - \text{Sim}_2(d_i, d_0))]. \tag{2}$$

Given that $d_0$, $d_1$, $d_2$, and $d_3$ are positioned on the same side relative to $q$, and assuming both $\text{Sim}_1$ and $\text{Sim}_2$ denote cosine similarity, let $\theta$ represent the angle between $d_0$ and $q$, and $x$ denote the angle between $d_i$ (i.e., $d_1, d_2, d_3$) and $d_0$. Thus, we get the Eq.3

$$\text{MMR} = \arg \max_{i=1,2,3} [0.5 \cdot (\cos(d_i, q) - \cos(d_i, d_0))] = \arg \max_{i=1,2,3} [0.5 \cdot (\cos(x + \theta) - \cos(x))] \tag{3}$$

The function $f(x) = \cos(x + \theta) - \cos(x)$, with its derivative $f'(x) = -\sin(x + \theta) + \sin(x)$, assumes $x$ and $x + \theta$ lie within $(0, \pi/2)$. Consequently, $f'(x) < 0$, indicating that for vectors on the same side of $q$, their MMR values decrease as the angle with $q$ increases. Thus, following the selection of $d_0$, the subsequent choices are $d_1$, then $d_2$, and so on. This sequence suggests that relevance predominantly influences the selection outcome.

The real challenge in vector retrieval emerges when $\lambda \neq 0.5$. The selection among candidate vectors $d_1$, $d_2$, and $d_3$ hinges critically on both $\lambda$ and $\theta$, complicating the determination of the most appropriate candidate. This dependency means that different query vectors and the distribution of initial candidate vectors require varying $\lambda$ values to achieve optimal performance. Consequently, it is impractical to predict the value of $\lambda$ in advance or to ascertain a precise direction for optimization. This issue becomes even more pronounced in higher-dimensional vector spaces, where the perturbations induced by changing $\lambda$ complicate the identification of an optimal adjustment direction. This inherent complexity underscores the need for adaptive retrieval strategies that dynamically adjust $\lambda$ based on the characteristics of the query and candidate vector distributions.

## 2.2 Sum Vector for Retrieval

To address the challenges associated with MMR, we propose utilizing the sum vector of selected vectors to simultaneously capture both similarity and diversity. Figure.1 provides an intuitive geometric illustration of vector diversity and similarity. In practical applications, similarity does not necessarily imply strict alignment between vectors; rather, a small angle between vectors suffices to indicate similarity. A smaller angle between two vectors denotes a higher degree of similarity, while a larger angle signifies lower similarity. In the context of an embedded query, retrieval typically involves ranking candidate vectors based on the angles they form with the query vector, from smallest to largest, and selecting them according to the required quantity, for instance, by directly applying cosine similarity. However, if the selected batch of vectors for a query includes vectors with both small and large angles relative to the query vector, the batch can be considered to satisfy the diversity requirement, as it incorporates vectors that are less directly relevant. Consequently, the selected vectors do not merely satisfy the criterion of having the smallest possible angles with the query vector but instead exhibit a range of angles. Nevertheless, diversity in vector retrieval should be grounded in similarity and should not entirely deviate from it. In other words, diversity should not be achieved by deliberately selecting vectors that are entirely different in angle from the query vector. The selected examples should exhibit substantial differences while satisfying the similarity constraint, thereby complementing each other and fulfilling both similarity and diversity requirements. And Sum vector can achieve this goal well. The sum vector vector of the selected vectors is required to closely approximate the query vector, thereby effectively satisfying the similarity constraint. While from a geometric perspective, the sum vector's similarity to the query vector implies that the selected vectors approach the query vector from different directions, indicating significant differences among the selected vectors while maintaining complementarity, which reflects diversity. Therefore, the sum vector effectively captures both diversity and similarity.

## 3 Vectors retrieval with similarity and diversity

### 3.1 Problem definition and complexity analysis

To address the problem of selecting a subset of vectors from a set of candidate vectors that satisfy both similarity and diversity requirements, we refer to the MMR algorithm and several LLM-based algorithms, typically considering the following premises: Firstly, the candidate vectors are identified from the entire set of vectors ($size = N$) using cosine similarity metrics, resulting in a subset of vectors ($size = n$). Consequently, this set of candidate vectors inherently exhibits a relative high degree of similarity to the query vector. Secondly, within these $n$ candidate vectors, the vector most similar to the query vector is typically selected first, as is the case with the MMR algorithm and others. This approach is favored because, in applications such as in-context learning with LLMs, examples with the highest similarity to the query are generally the most helpful.

As previously mentioned, while algorithms like MMR are widely applied in practice, these studies often lack a robust and reliable theoretical model. In other words, many approaches employ heuristic strategies or machine learning methods to arrive at a solution without providing a rigorous formal description and analysis of similarity and diversity from a theoretical perspective. Therefore, based on the aforementioned premises, we propose using the sum vector to characterize both similarity and diversity in vector retrieval. The definition of the sum vector is as follows:

**Definition 1.** *The Sum Vector: Given $k$ vectors $d_1, d_2, ..., d_k$, the sum vector $d$ is the sum of these $k$ vectors.*

Specifically, we aim to maximize the similarity between the sum vector of the selected $k$ vectors and the query vector. On one hand, maximizing the similarity of the sum vector to the query vector imposes a similarity constraint. On the other hand, from a geometric perspective, ensuring the sum vector is similar to the query vector means that the selected vectors approach the query vector from different directions, thus imposing a diversity constraint. Therefore, using the sum vector to characterize similarity and diversity allows us to model complex semantic similarity and diversity through simple vector addition operations. Next, we define the problem of vectors retrieval as follows:

**Definition 2.** *The problem of Vectors Retrieval with Similarity and Diversity (VRSD): Given a query vector $q$ and a set of candidate vectors $R = \{d_0, d_1, ..., d_{n-1}\}$ (where $d_0$ is the vector with the*

*highest similarity to query vertor q), $d_0$ is selected first because of its highest similarity. Then, how to select $k - 1$ vectors $(d'_1, d'_2, ..., d'_{k-1})$ from the remaining vectors such that the cosine similarity between the sum vector $d = d_0 + d'_1 + d'_2 + ... + d'_{k-1}$ and $q$ is maximized.*

The vector $d_0$, characterized by its maximal similarity to the query vector $q$, establishes an initial constraint on similarity. The ensuing optimization objective strives to maximize the cosine similarity between the sum vector of all selected vectors and $q$. This process necessitates the selection of vectors that not only converge towards $q$ from diverse dimensions but also exhibit significant diversity and complementarity. However, upon further examination of above problem, we find that it is an NP-complete problem. Below, we provide a theoretical proof. Since the vector $d_0$, with the highest similarity, is initially selected, the subsequent selection of $k - 1$ vectors must have the maximum cosine similarity with $q - d_0$. That is, maximizing the similarity between sum vector $d = d_0 + d'_1 + d'_2 + ... + d'_{k-1}$ ($d'_1, d'_2, ..., d'_{k-1}$ represents the $k - 1$ vectors selected subsequently) and $q$, is equivalent to maximizing the similarity between $d' = d'_1 + d'_2 + ... + d'_{k-1}$ and $q - d_0$. To this end, we define a decision problem, namely:

**Definition 3.** *The decision problem of vectors retrieval: Given a set of candidate vectors $R$ and a query vector $q$, can $k$ vectors be selected from $R$ such that the cosine similarity between the sum vector of these $k$ vectors and the query vector $q$ equals 1? We denote instances of this vectors retrieval problem as $(R, q, k)$.*

Next, we will prove this decision problem is NP-complete. For the sake of concise proof, we further restrict the components of vectors to integers. The proof strategy is to reduce the subset sum problem Cormen et al. (2009) to this decision problem.

**Definition 4.** *The subset sum problem: Given an integer set $T$ and another integer $t$, does there exist a non-empty subset whose sum of elements equals $t$? We denote instances of the subset sum problem as $(T, t)$.*

For the convenience of proof, we also need to define a modified version of the subset sum problem, called the $k$-subset sum problem.

**Definition 5.** *$k$-subset sum problem: Given an integer set $T$ and another integer $t$, does there exist a non-empty subset of size $k$ (i.e., the cardinality of the subset is $k$), whose sum of elements equals $t$? We denote instances of the $k$-subset sum problem as $(T, t, k)$.*

**Lemma 1.** *The $k$-subset sum problem is NP-complete.*

*Proof.* We reduce the subset sum problem(Def.4) to the $k$-subset sum problem(Def.5) .

1. Clearly, the $k$-subset sum problem is polynomial-time verifiable.

2. Reducing the subset sum problem to the $k$-subset sum problem.

For any instance of the subset sum problem $(T, t)$, we can transform it into $|T|$ instances of the $k$-subset sum problem, i.e., $(T, t, 1), (T, t, 2), ..., (T, t, |T|)$. If any of these $|T|$ instances of the $k$-subset sum problem has a yes answer, then the answer to the subset sum problem is yes. If all answers to these $|T|$ instances of the $k$-subset sum problem are no, then the answer to the subset sum problem is also no. Therefore, if the $k$-subset sum problem can be solved in polynomial time, then the subset sum problem can also be solved in polynomial time. Hence, the $k$-subset sum problem is NP-complete. $\square$

Now it is time to prove the NP-completeness of the decision problem of vectors retrieval.

**Theorem 1.** *The decision problem of vectors retrieval is NP-complete.*

*Proof.* We reduce the $k$-subset sum problem(Def.5) to the decision problem of vectors retrieval(Def.3).

1. The answer to vectors retrieval is polynomial-time verifiable. If the answer provides $k$ vectors, we can simply add these $k$ vectors and then calculate whether the cosine similarity between the sum vector and the query vector $q$ equals 1. This verification can be done in polynomial time.

2. Reducing the $k$-subset sum problem to the decision problem of vectors retrieval.

For any instance of the $k$-subset sum problem $(T, t, k)$, let $T = \{t_1, t_2, \ldots, t_n\}$. We construct the set of vectors $R$ and the query vector $q$ as Eq.4:

$$R = \{[t_1, 1], [t_2, 1], \ldots, [t_n, 1]\}, q = [t, k] \tag{4}$$

The decision problem of vectors retrieval $(R, q, k)$ asks whether there exist $k$ vectors such that the sum vector (denoted as $d$) of these vectors and the query vector $q$ have a cosine similarity of 1. According to the definition of cosine similarity, $cos\_similarity = \frac{d \cdot q}{|d| \cdot |q|}$. The cosine similarity between $d$ and $q$ equals 1 if and only if $d = \alpha q$, where $\alpha$ is a constant. Therefore, if vectors retrieval provides an affirmative answer $d = \alpha q$, we can get the Eq.5,

$$d = [t'_1, 1] + [t'_2, 1] + \ldots + [t'_k, 1] = \alpha[t, k] \Rightarrow [(t'_1 + \ldots + t'_k), k] = \alpha[t, k]. \tag{5}$$

$[t'_1, 1] \ldots [t'_k, 1]$ are the selected $k$ vectors. It implies that $\alpha = 1$ and $t'_1 + \ldots + t'_k = t$. Thus, this provides an affirmative answer to the $k$-subset sum problem instance $(T, t, k)$. Conversely, if vectors retrieval provides a negative answer, then a negative answer to the $k$-subset sum problem can also be obtained. The above reduction process can be clearly completed in polynomial time. **Therefore, the decision problem of vectors retrieval is NP-complete.** $\square$

## 3.2 HEURISTIC ALGORITHM FOR VECTORS RETRIEVAL

Since the vectors retrieval problem $(R, q, k)$ is a NP-complete problem, necessitating the use of heuristic methods to derive feasible solutions. Specifically, given a set of candidate vectors with high similarity, the objective is to select $k$ vectors that maximize the cosine similarity between the sum vector of the $k$ selected vectors and the query vector. We propose a new algorithm denoted as **V**ectors **R**etrieval with **S**imilarity and **D**iversity (**VRSD**). VRSD initially selects the vector most similar to the query vector and then iteratively selects additional vectors from the remaining candidates. In each iteration, it chooses the vector that maximizes the cosine similarity between the cumulative sum of all previously selected vectors and the query vector, continuing this process until $k$ vectors are chosen. Further details about the VRSD algorithm can be found in Algorithm.1.

---

**Algorithm 1** Vectors Retrieval with Similarity and Diversity (VRSD)

---

**Require:** Candidate vector set $R = \{d_0, d_1, \ldots, d_{n-1}\}$, query vector $q$, where $d_0$ is the vector from all $d_i$ that has the highest cosine similarity with $q$, and constant $k$.
**Ensure:** $k$ vectors including $d_0$, such that the cosine similarity between the sum vector of these $k$ vectors and $q$ is maximized.
1: $S = \{d_0\}$
2: $R = R \setminus \{d_0\}$
3: **for** $i = 1$ to $k - 1$ **do**
4:     $s = \sum S$                                           $\triangleright$ Sum of all vectors in $S$
5:     maxCos $= -1$
6:     $p = $ null                           $\triangleright$ Initialize $p$ to a null vector or equivalent
7:     **for** $v$ in $R$ **do**
8:         $t = s + v$                         $\triangleright$ Temporary vector for comparison
9:         **if** $\cos(t, q) > $ maxCos **then**
10:             maxCos $= \cos(t, q)$
11:             $p = v$
12:         **end if**
13:     **end for**
14:     $S = S \cup \{p\}$                                  $\triangleright$ Add $p$ to the set $S$
15:     $R = R \setminus \{p\}$                                $\triangleright$ Remove $p$ from $R$
16: **end for**
17: return $S$

---

## 3.3 TIME COMPLEXITY ANALYSIS OF VRSD

As depicted in Algorithm.1, the time complexity of the VRSD algorithm is $k \times |R| = k \times n$, which accounts for the initial step of selecting $n$ candidate vectors from the entire set of vectors (size =

$N$) based on similarity. Given that $N \gg n > k$, the computational load of subsequent steps in Algorithm.1 is minimal in comparison. The MMR algorithm, which also selects $k$ vectors from $|R|$ candidates, requires two iterations of maximum calculations as depicted in Eq.1—once for each candidate vector against the query vector and once against the set of already selected vectors $|S|$. Thus, the complexity for MMR becomes $k \times |R| \times |S| = k \times |R|^2 = k \times n^2$, indicating a marginally higher computational demand compared to VRSD.

## 4 EXPERIMENTS

### 4.1 IMPLEMENTATION DETAILS

We evaluated the VRSD algorithm using three publicly available datasets of different categories and compared the VRSD with the MMR algorithm when the values of $\lambda$ are 0, 0.5, and 1 respectively :

- **ARC-DA** (Bhakthavatsalam et al., 2021): A dataset of direct-answer science questions derived from the ARC multiple-choice question. Each example contains a question and multiple answers.
- **OpenBookQA** (Mihaylov et al., 2018): A dataset of multiple-choice science questions, which probe the understanding of science facts and the application of these facts to novel situations. Each example contains a question, multiple choices, and an answer.
- **Puzzle** (Liu et al., 2023): A question answering dataset. These questions belong to lateral thinking puzzle. Each example contains a question and an answer.

For each item in each datasets, we concatenate the question part with its corresponding answer, subsequently selecting 20% of these concatenated items to form the test set, wherein the question parts are isolated. Items designated for the test set are excluded from the original dataset for subsequent experiments, where same amount of examples are retrieved for each test question.

As extensively discussed in Section 2, sum vector can well capture the similarity and diversity simultaneously. Consequently, retrieval quality is assessed by aggregating all vectors retrieved using either VRSD or MMR into a sum vector—denoted as $d_{\text{VRSD}}$ and $d_{\text{MMR}}$—which reflects the vectorial direction from which the examples approach the query vector $q$. We compute the cosine similarity between the sum vectors and the query vector as $\cos(d_{\text{VRSD}}, q)$ and $\cos(d_{\text{MMR}}, q)$. The comparison includes counting percentage where $\cos(d_{\text{VRSD}}, q)$ exceeds $\cos(d_{\text{MMR}}, q)$, termed as the **VRSD win. rate**, and calculating the maximum difference (**Max-diff**) between these cosine similarities for all queries in each test set. Additionally, we compute the mean cosine similarity values (**Mean**) for these vectors under each methods, respectively. Such metrics are instrumental in elucidating the algorithms' capacity to balance similarity and diversity. The comparison was conducted on one open source model all-mpnet-base-v2 and two close source models text-embedding-3-small&text-embedding-ada-002.

The author of MMR conducted a manual evaluation of the retrieved examples, noting that 'users were asked to extract information from documents without being informed about the order in which the documents were presented—only that either "method R" or "method S" was applied.' (Carbonell & Goldstein, 1998). Manual evaluation, however, is time-consuming and labor-intensive. We propose utilizing the sum vector for evaluation, which offers a more direct and efficient approach. As discussed earlier, the sum vector can simultaneously account for both diversity and similarity.

### 4.2 EXPERIMENTAL RESULTS

#### 4.2.1 RETRIEVAL PERFORMANCE

Table.1 presents the overall result of MMR and VRSD on ARC-DA, OpenBookQA and Puzzle. From the results, we have the following observations and conclusions:

We observe that the win rate of VRSD consistently exceeds 90% compared to MMR across various datasets and conditions. This suggests that, without requiring additional parameters, VRSD retrieves examples that are more pertinent to the original query and more effectively satisfies diversity requirements. Notably, VRSD maintains an advantage over MMR in all scenarios concerning the Max-diff, underscoring VRSD's superior capacity to leverage vector information for more diverse

Table 1: Results of MMR and VRSD on three datasets. The best results are bolded.

| Model | Algorithm | ARC-DA | | | OpenBookQA | | | Puzzle | | |
|---|---|---|---|---|---|---|---|---|---|---|
| | | VRSD win.rate | Max-diff | Mean | VRSD win.rate | Max-diff | Mean | VRSD win.rate | Max-diff | Mean |
| all-mpnet-base-v2 | VRSD | - | - | **0.740** | - | - | **0.833** | - | - | **0.592** |
| | MMR($\lambda$=0) | 97.7% | 0.160 | 0.696 | 97.3% | 0.135 | 0.809 | 100% | 0.161 | 0.537 |
| | MMR($\lambda$=0.5) | 92.5% | 0.108 | 0.720 | 92.6% | 0.101 | 0.822 | 90% | 0.052 | 0.576 |
| | MMR($\lambda$=1) | 95.3% | 0.158 | 0.710 | 96.8% | 0.102 | 0.812 | 100% | 0.132 | 0.577 |
| text-embedding-3-small | VRSD | - | - | **0.695** | - | - | **0.751** | - | - | **0.572** |
| | MMR($\lambda$=0) | 98.8% | 0.145 | 0.652 | 98.1% | 0.137 | 0.724 | 100% | 0.184 | 0.555 |
| | MMR($\lambda$=0.5) | 92.3% | 0.113 | 0.676 | 93.3% | 0.130 | 0.738 | 90% | 0.052 | 0.527 |
| | MMR($\lambda$=1) | 96.5% | 0.129 | 0.669 | 97.3% | 0.085 | 0.729 | 100% | 0.132 | 0.518 |
| text-embedding-ada-002 | VRSD | - | - | **0.905** | - | - | **0.922** | - | - | **0.893** |
| | MMR($\lambda$=0) | 98.5% | 0.040 | 0.895 | 97.2% | 0.043 | 0.915 | 100% | 0.025 | 0.881 |
| | MMR($\lambda$=0.5) | 90.8% | 0.027 | 0.901 | 88.1% | 0.032 | 0.918 | 95% | 0.020 | 0.888 |
| | MMR($\lambda$=1) | 95.0% | 0.041 | 0.897 | 95.0% | 0.028 | 0.915 | 95% | 0.018 | 0.887 |

Table 2: The performance of retrieved examples under each method with different LLMs. We calculate the standard error of the mean of each value. The best results are bolded.

| Algorithm | ARC-DA | | OpenBookQA | | Puzzle | |
|---|---|---|---|---|---|---|
| | gpt-3.5-turbo | open-mistral-7b | gpt-3.5-turbo | open-mistral-7b | gpt-3.5-turbo | open-mistral-7b |
| VRSD | **0.371 ± 0.008** | **0.233 ± 0.006** | **0.789 ± 0.004** | **0.534 ± 0.008** | **0.213 ± 0.007** | **0.198 ± 0.018** |
| MMR($\lambda$=0) | 0.355 ± 0.010 | 0.216 ± 0.004 | 0.767 ± 0.012 | 0.508 ± 0.014 | 0.206 ± 0.002 | 0.198 ± 0.015 |
| MMR($\lambda$=0.5) | 0.364 ± 0.011 | 0.218 ± 0.008 | 0.772 ± 0.006 | 0.507 ± 0.014 | 0.202 ± 0.002 | 0.188 ± 0.012 |
| MMR($\lambda$=1) | 0.347 ± 0.013 | 0.222 ± 0.005 | 0.780 ± 0.006 | 0.510 ± 0.014 | 0.188 ± 0.002 | 0.186 ± 0.011 |

and relevant retrieval. Furthermore, MMR consistently underperforms across all datasets in terms of the Mean, which measures the average cosine similarity between the sum vector and the query vector. This finding indicates that the overall effectiveness of MMR retrieval is generally inferior to that of VRSD, and this discrepancy is not incidental. Additionally, the significant difference in Mean between VRSD and MMR suggests that the two methods capture different vectors when retrieving passages, further highlighting the distinction in their retrieval mechanisms.

### 4.2.2 DOWNSTREAM TASK EXECUTION PERFORMANCE

To verify that the improvements of VRSD in retrieval, with respect to both similarity and diversity, effectively enhance downstream tasks, we conducted a validation study using two models: the open-source LLM, Open-Mistral-7b, and the closed-source LLM, Gpt-3.5-Turbo. In this study, we reconstructed prompts by concatenating the original instances corresponding to the retrieved vectors with the initial query, and then input these prompts into the LLMs. The responses generated by the LLMs, based on different retrieval algorithms, were compared with standard answers using the ROUGE-L (Lin, 2004) metric for the ARC-DA and Puzzle datasets, and Accuracy (Schütze et al., 2008) for the OpenBookQA dataset, respectively. This evaluation allowed us to assess the efficacy of the retrieved examples in facilitating accurate responses. Furthermore, given that VRSD consistently outperformed MMR in retrieval performance across all models and datasets in previous results, we selected the model all-mpnet-base-v2 for retrieval prior to executing the downstream tasks.

The results in Table.2 shows that VRSD achieves the highest scores across all metrics for answer quality across all datasets. This suggests that the examples retrieved by VRSD better enhance the LLM's understanding of the query and facilitate the generation of more accurate answers. Additionally, GPT-3.5-turbo outperforms open-mistral-7b, demonstrating that models with superior instruction-following capabilities benefit more from diverse and relevant examples. Moreover, as shown in Table.1, VRSD produces superior retrieval results, and these examples, when used by LLMs, lead to improved task performance. This not only confirms that the sum vector more effectively captures both diversity and similarity, but also that the retrieved examples substantially enhance the reliability of the LLM's answers.

Overall, VRSD demonstrates superior performance compared to MMR in both retrieval effectiveness and task execution, meeting the demands of both similarity and diversity without requiring parameter adjustments. This underscores the distinct advantages of VRSD. Notably, in the Puzzle dataset, the mean values of $\cos(d_{\text{VRSD}}, q)$ and $\cos(d_{\text{MMR}}, q)$ are comparatively low and the discrepancy between open-source and closed-source models is less significant, with VRSD achieving a 100% win rate, likely due to the small dataset size. Nonetheless, since lateral puzzle questions necessitate that LLMs grasp the query and generate insights from multiple perspectives in the retrieved examples, the Puzzle dataset remains an important benchmark for evaluating our algorithm.

Table 3: Performance comparision of different algorithms on datasets. For each dataset, The first three indicators assess retrieval performance, while the latter two evaluate task execution performance. **Win** denotes VRSD win.rate.

| Algorithm | ARC-DA | | | | | OpenBookQA | | | | | Puzzle | | | | |
|---|---|---|---|---|---|---|---|---|---|---|---|---|---|---|---|
| | Win | Max-diff | Mean | GPT-3.5-turbo | open-mistral-7b | Win | Max-diff | Mean | GPT-3.5-turbo | open-mistral-7b | Win | Max-diff | Mean | GPT-3.5-turbo | open-mistral-7b |
| VRSD | - | - | 0.740$^{\dagger\dagger}$ | 0.371$^{\dagger\dagger}$ | 0.233$^{\dagger\dagger}$ | - | - | 0.833$^{\dagger\dagger}$ | 0.789$^{\dagger\dagger}$ | 0.534$^{\dagger\dagger}$ | - | - | 0.592$^{\dagger\dagger}$ | 0.213$^{\dagger\dagger}$ | 0.198$^{\dagger\dagger}$ |
| CS | 95.3% | 0.158 | 0.710$^{\dagger}$ | 0.361$^{\dagger}$ | 0.222$^{\dagger}$ | 96.8% | 0.102 | 0.812$^{\dagger}$ | 0.780$^{\dagger}$ | 0.510$^{\dagger}$ | 100% | 0.132 | 0.577$^{\dagger}$ | 0.201$^{\dagger}$ | 0.187$^{\dagger}$ |
| BM25 | 98.7& | 0.684 | 0.550 | 0.347 | 0.196 | 99.9% | 0.650 | 0.544 | 0.753 | 0.503 | 100% | 0.262 | 0.459 | 0.188 | 0.185 |

## 4.3 FURTHER ANALYSIS

Next, we perform a comprehensive analysis of BM25 and Cosine Similarity across all datasets to evaluate their performance in terms of diversity and similarity in retrieval. As BM25 and Cosine Similarity are classical distance measurement methods, it is crucial to assess their performance in relation to our sum vector; this enables us to understand their behavior within the context of the sum vector and justifies their inclusion in our comparison. While both BM25 and Cosine Similarity are rank-based retrieval approaches, VRSD distinguishes itself by simultaneously capturing both diversity and similarity, retrieving all relevant examples in one step and considering the potential connections between embedded vectors. For consistency, we select the same number of top-ranked examples for evaluation, similar to previous settings. As in prior experiments, we utilize the all-mpnet-base-v2 model for retrieval before executing downstream tasks.

Table.3 illustrates a notable performance decline when employing BM25 or direct cosine similarity for retrieval in terms of diversity and similarity, thereby confirming the superiority of VRSD. Furthermore, as reflected in the Mean metric, the retrieval performance follows the order: VRSD > Cosine Similarity > BM25. Similarly, the responses from both LLMs demonstrate the same downward trend. This finding suggests that VRSD more effectively balances diversity and relevance in the retrieved examples, ultimately enhancing the quality of LLM-generated responses. These insights from downstream tasks provide further evidence supporting our earlier assertion that the sum vector implicitly captures the diversity of the set while also accounting for similarity.

## 5 RELATED WORK

Retrieval methods in Large Language Models (LLMs) have gained traction, particularly due to their pivotal role in open-domain question answering, evidenced by seminal contributions in the field (Chen et al., 2017; Guu et al., 2020; Khattab & Zaharia, 2020; Qu et al., 2021; Izacard & Grave, 2021). The introduction of Retrieval Augmented Generation (RAG) further underscored the significance of these methods across knowledge-intensive tasks (Lewis et al., 2020), notably enhancing generation capabilities for open-domain queries (Mao et al., 2021). This spurred the adoption of techniques such as K-Nearest-Neighbor (KNN) in diverse applications, ranging from the customization of multilingual models in machine translation (Khandelwal et al., 2020) to improving the prediction of rare patterns in LLMs (Khandelwal et al., 2019; Alon et al., 2022). Continued advancements in retrieval techniques have focused on identifying highly informative examples to augment in-context learning, thereby enabling LLM-based systems to achieve significant performance improvements with minimal examples (Brown et al., 2020). Early on, traditional sparse retrieval methods like BM25 (Robertson et al., 2009)—an extension of TF-IDF—were utilized to refine in-context learning (Liu et al., 2022). Subsequently, the integration of LLMs' intrinsic capabilities (Shin et al., 2021) and Sentence-BERT (SBERT) (Reimers & Gurevych, 2019) facilitated the retrieval of highly pertinent examples for prompt integration. The advent of dense retrievers signified a methodological enhancement in retrieval from a machine learning perspective, and the incorporation of feedback signals with contrastive learning has yielded more effective retrieval systems (Rubin et al., 2022; Wang et al., 2023). Recent innovations like UPRISE (Cheng et al., 2023) and PRAC (Nie et al., 2023) have further optimized the performance of in-context learning by retrieving demonstrations directly from training data. Despite these advances, most retrieval methods still treat each candidate independently, which can lead to suboptimal outcomes due to the interaction effects among in-context examples, resulting in a lack of diversity. Besides, given the expanding applications of LLMs, diversity becomes increasingly crucial, incorporating a diverse range of examples enriches LLMs' learning processes, facilitating more innovative and robust responses, especially for complex open-ended questions. Most existing methods in this area rely on MMR; however, the necessity for

continuous adjustment to determine the optimal value of $\lambda$ is inefficient in practical applications (Carbonell & Goldstein, 1998; Deselaers et al., 2009; Ye et al., 2023). In this work, we aim to retrieve examples from the perspective of combinatorial optimization, supported by strict theoretical analysis. Furthermore, the proposed sum vector has been demonstrated to effectively capture both diversity and similarity, contributing to the successful execution of downstream tasks.

## 6 CONCLUSIONS

In this work, considering the complexity of parameter adjustment in MMR, we aim to improve how LLMs retrieve similar yet diverse examples by introducing a novel approach that jointly models both constraints through the relationship between the sum vector and the query vector. This method ensures that individual vectors within the sum diverge from the query vector, thereby satisfying the diversity constraint. We demonstrate that this problem is NP-complete and propose the VRSD algorithm, which not only surpasses MMR in retrieval performance but also enhances the execution of downstream tasks. Additionally, we introduce the sum vector, which is proven to effectively capture both diversity and similarity simultaneously, and serves as a metric to measure the balance between these two aspects, addressing an existing gap. Our work highlights the inherent challenges of achieving both similarity and diversity in vector retrieval, establishing a solid theoretical foundation for future research. The proposed combinatorial optimization problem holds independent value from both theoretical and practical perspectives, suggesting that further exploration and refinement of the heuristic algorithm would be a fruitful direction for future inquiry.

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
