# OpenReview forum: "VRSD: Rethinking Similarity and Diversity for Retrieval in Large Language Models"
_ICLR.cc/2025/Conference — ICLR 2025 Conference Withdrawn Submission_

### Official Review · Reviewer_j7XU · 2024-10-26

**Soundness:** 1
**Presentation:** 2
**Contribution:** 2
**Rating:** 3
**Confidence:** 4

**Summary:**

The paper analyzes a commonly used method for balancing diversity and relevance, Maximal Marginal Relevance (MMR). The authors propose a new method that uses sum vector (VRSD) as a single metric to balance the two aspects, present theoretical analyses for their approach and a greedy method to implement VRSD in practice. The authors evaluate their approach on top of LLM-generated embeddings in both pure retrieval setting (wrt sum vector as metric) and in end-to-end QA settings.

**Strengths:**

* Balancing relevancy and diversity is an important problem for IR, emerging use cases like RAG, etc. Existing approaches including MMR DPP are either complex to implement (eg O(N^3) complexity in DPP) or rely on some heuristics (MMR).

* Developing simpler proxy metrics (eg potentially sum vector-based similarity) can be valuable to improve progress.

**Weaknesses:**

* Formulation. It's unclear if sum vector-based similarity is a desirable metric to balance diversity and relevance to begin with. Let $f$ be a function that maps a set of candidate items $X$ to a scalar $r \in \mathbb{R}$. The ideal function should be monotonic in the sense that as more elements are added to the set $X$, $r$ should be monotonic increasing (ie adding more candidates should not hurt the overall retrieval stage's quality). cos(sum vector, query vector) as proposed in this paper doesn't satisfy this property.

* Evaluation. Given a new metric balancing similarity and diversity is proposed, extensive evaluations should be done. But there are significant weaknesses in the existing evaluations. Specifically,

    *  Section 4.1 starts out by treating sum vector as ground truth (without any manual evaluation on the quality) (per "Consequently, retrieval quality is assessed by aggregating all vectors retrieved using either VRSD or MMR into a sum vector"). This is then used to justify superiority of (greedy-based) VRSD over MMR.
        * First, in IR, performing manual evaluation is critical to justify if diversity-based metrics work. This is done in many prior work, not just MMR.
        * Second, given sum vector is treated as ground truth, isn't greedy-based VRSD winning over MMR expected as the latter doesn't optimize for the desired metrics?

    * No grid search over hyperparameter in MMR. I would expect grid search in 0.1 … 0.9 at the very least.

    * Section 4.2.2, Section 4.3,  Table 3. I don't understand the logic behind experiment construction here. It's fine to say "greedy based VRSD" is better than MMR for optimizing VRSD metric (Sec 4.1), but how does the result in Sec 4.1 justify leaving out MMR-based approaches (and other alternatives like DPP, \alpha-NDCG, etc.) altogether in this section?  The baselines like BM25 and Cosine Similarity don't capture diversity at all.  So aren't we really showing "method X that capture notion of diversity is better than other methods that don't capture any diversity" in these two sections?

* Related work. Should probably discuss and/or compare with alternative approaches/metrics like $\alpha$-NDCG and Determinantal Point Process in addition to MMR.

* Misc typos: 347: in each datasets -> dataset

**Questions:**

Please see W section. The experiment flows should be cleaned up as the current logic is confusing (eg dropping MMR for the second group of experiments), and I feel more theoretical analysis (eg why the lack of monotonicity in the proposed metric is not problematic) and empirical analysis (eg manual eval, grid search with more hyperparameters and comparison with alternative approaches like DPP) would be extremely helpful.

---

### Official Review · Reviewer_JNmj · 2024-11-04

**Soundness:** 2
**Presentation:** 1
**Contribution:** 2
**Rating:** 3
**Confidence:** 4

**Summary:**

The paper tackles the problem of combining computation of the relevance and diversity scores using the cosine similarity between a query and a sum vector of top-K candidates. Diversity is an important problem in information retrieval, but I have not seen many recent works, so revisiting the diversity problem is clearly beneficial.

Authors argue that the proposed approach solves the problem with the well-known maximum marginal relevance (MMR) method whose performance depends greatly (as claimed by the authors) on the parameter lambda. They present a proof that selecting a subset of vectors that maximizes the sum-vector cosine similarity with the query is NP-complete and, thus, solve the problem approximately using a (seemingly greedy) approach.

Moreover, they evaluate performance of the method using intrinsic and extrinsic evaluation using the sum-vector for QA tasks in seemingly retrieval-augmented-generation (RAG) scenario and find it to be superior to MMR with lambda in {0, 0.5, 1}.

I think it can be potentially an interesting paper. I like how authors used RAG to show the usefulness of their approach. Yet, I think it is largely unreadable in the present form (some examples/questions are shown in the detailed notes) and I do not think it is possible to resolve this problem just for a camera ready version. Moreover, it has issues related to using only a single dense-retrieval model (**in the downstream task**) and using potentially inappropriate metric for intrinsic evaluation.

As a somewhat minor, but still non-negligible issue: the reduction to an NP-complete problem with a pseudo-polynomial solution is not sufficient to show that the problem has no efficient exact solutions (and thus a greedy approximation is required).

**Strengths:**

* The paper revisits an important and often neglected problem of diversity in information retrieval.
* They propose a simple heuristic approach, which outperforms prior art MMR in their evaluations.
* The evaluation includes downstream generative QA task

**Weaknesses:**

* The paper is **VERY** hard to read and omits important details as well as some citations for recent work.
* The intrinsic evaluation approach uses the metric that was not properly justified (and likely it is not a good one)
* Only a single dense retrieval model was used in the downstream task evaluation.
*  The reduction to an NP-complete problem with a pseudo-polynomial solution is not sufficient to show that the problem has no efficient exact solutions (and thus a greedy approximation is required).

**Questions:**

All of these are suggestions/detailed comments (not questions)

1. Section 2,1 Limitations of MMR:
* First of all, you do not explain that MMR is an interactive process. Without this Eq. (1) is very hard to understand.
* Second, I read your example twice, but I am convinced that it does NOT explain the limitations of MMR. I cannot put a finger on what is exactly wrong in this narrative, but it IMHO lacks a clear a concise explanation where you have an AHA-moment in the end. In particular, in L159-161 you write: "The selection among candidate vectors d1, d2, and d3 hinges critically on both λ and θ, complicating the determination of the most appropriate candidate. This dependency means that different query vectors and the distribution of initial candidate vectors require varying λ values to achieve optimal performance."

To make this statement you do **NOT** need any example at all, it is quite clear that lambda needs to be tuned. That said, in  your example, it is not clear why one or the other order of candidates should be preferred. If you were able to make such statements, than you would have been able to show that changing the value of lambda changes the order in very undesirable ways.

2. It is IMHO not enough to show NP-completeness by reduction to the integer k-sum problem, because this problem has an efficient pseudo-polynomial time dynamic-programing (DP) solution. To reiterate, it is NP-complete, but it does not mean much in many practical cases. However, the DP is enabled by the "additivity" of the problem. In contrast, the maximum cosine problem with a sum-vector is not additive due to multi-dimensionality, so I think there is no pseudo-polynomial solution. To reiterate the actually useful proof, we have to show that the problem has no easy solutions such as DP.

3. The intrinsic evaluation just counts "percentage where cos(dVRSD, q) exceeds cos(dMMR, q), termed as the VRSD win." It is not clear to me that comparing these cosines is meaningful. To rephrase, why having lower cosines entails higher diversity? I do not think so, but there are better ways to measure diversity, which may include even running additional human evaluations.

Other suggestions to improve intrinsic evaluations (these are only recommendations though, I don't have a strong opinion what is the right approach, but I am convinced just comparing cosines is **a wrong one**):
i. Authors could consider some standard benchmarks such as diversity task from TREC Web track: Clarke, Charles LA, Nick Craswell, Ian Soboroff, and Ellen M. Voorhees. "Overview of the TREC 2011 Web Track." In TREC. 2011.
ii. Min et al recently measured diversity in QA tasks by an ability to retrieve diverse set of answers: Joint Passage Ranking for Diverse Multi-Answer Retrieval, Min et al. 2021.
iii. It maybe worth looking at the following benchmark: Vikraman, Lakshmi, et al. "Passage similarity and diversification in non-factoid question answering." Proceedings of the 2021 ACM SIGIR International Conference on Theory of Information Retrieval. 2021.

4. Experimental description section. This section largely lacks structure. As one prominent example:

L347-349 For each item in each datasets, we concatenate the question part with its corresponding answer, subsequently selecting 20% of these concatenated items to form the test set, wherein the question parts are isolated.

This seemingly describes spitting into the development/training and testing subsets. Ok, why do we mention concatenation of the question and the corresponding answer here? I have no idea.

L406 and the downstream eval section:

"In this study, we reconstructed prompts by concatenating the original instances corresponding to the retrieved vectors with the initial query, and then input these prompts into the LLMs."

* First, this is seemingly a very awkward way to describe RAG (a citation is needed), but you don't name it!
* Second, reconstruction seems to be a wrong word here. Reconstruction basically means fixing or repairing something broken/split, but you just construct a new prompt.

---

### Official Review · Reviewer_gFB8 · 2024-11-04

**Soundness:** 1
**Presentation:** 2
**Contribution:** 1
**Rating:** 3
**Confidence:** 4

**Summary:**

This paper tries to tackle the dual goals of vector retrieval in maximizing relevance while maintaining source diversity. It attempts to formulate diversity as a mathematical construct that can be optimized for, as opposed to human evaluation. While the approach seems appealing, the formulation seems suspect, leading to a fundamental flaw in the paper.

**Strengths:**

The paper is easy to read. It tackles an important problem in vector retrieval, which is lack of diversity. Vector retrieval can be especially prone to herding, where near duplicates are surfaced in an attempt to maximize relevance (similarity).

**Weaknesses:**

The fundamental weakness of the paper is at the heart of how the paper defines diversity through the sum vector. Because the paper treats this as self evident, everything including evaluation hinges on this definition.

The paper claims that diversity is maximized when the dot product (without loss of generality, we can assume vectors are unit normed, as they are in most vector retrieval applications) between the sum vector and the query vector is maximized. That is (sum_i d_i) . q_i. But trivially that is the same as sum_i (d_i . q_i). Therefore the dot product between the query vector and the sum vector is maximized when each vector is as similar to the query as possible!

The paper states without proof that maximizing the sum vector needs that the document vector needs to "approach the query vector from different dimensions". I'm not sure why this is the case.

Since the paper takes it as a given that the sum vector is the best definition of diversity, it also uses that as the metric, dismissing the human evaluation approach in MMR as "time consuming". On the contrary, human evaluation is the gold standard for measuring diversity of results. Perhaps including qualitative analysis would have surfaced some flaws in the approach. It's no wonder that the paper comes out ahead as the measurement of diversity is based on what the paper chooses to optimize.

**Questions:**

Why is it self evident that maximizing the sum vector maximizes diversity? Why is it not the case that the cosine similarity with the sum vector is maximized by setting each vector as close to q as possible?

Why would we not codify the diversity constraint in the objective? For example, we could minimize pairwise diversity d_i.d_j even while maximizing the relevance of each document d_i.q to the query.

---

### Official Review · Reviewer_Moa1 · 2024-11-06

**Soundness:** 3
**Presentation:** 3
**Contribution:** 3
**Rating:** 5
**Confidence:** 4

**Summary:**

The paper is concerned with the problem of retrieving relevant yet diverse results when querying large language models. To tackle this problem, authors introduce a new algorithm called VRSD (Vectors Retrieval with Similarity and Diversity) designed to retrieve vectors that simultaneously meet similarity and diversity criteria. VRSD outperforms the widely used MMR (Maximal Marginal Relevance) algorithm, which has the drawback of requiring parameter tuning to balance similarity and diversity.

The core idea behind VRSD is that it works with the sum vector of the documents. A sum vector which is close to the query vector is linked to similarity, while requiring individual vectors within the sum vector to diverge in their alignment with the query vector satisfies the diversity constraint.

The study also shows that VRSD enhances the performance of downstream tasks, such as response generation by LLMs (Large Language Models). The article compares VRSD with MMR, BM25, and Cosine Similarity across various datasets, and the results demonstrate that VRSD’s outperforms MMR.

**Strengths:**

The paper tackles a relevant issue and the proposal is intuitive and it seems to be effective, although just one baseline technique was employed. In detail, the authors propose a greedy heuristic that combines vector-space representations of both similarity and diversity. The proposal is validated experimentally using 3 publicly available datasets and outperformed the baselines. I do not consider the originality and significance outstanding (details below), given 60 years of works on vector-based models in information retrieval and recommender systems. On the other hand, the paper reads well and it is clear. The paper's quality with respect to both the theoretical discussion and experimental methodology is good.

**Weaknesses:**

The paper seems to have two main weaknesses as submitted.

First of all, I missed some characterization that grounds the theoretical arguments of the paper. Considering the datasets employed, is there really a range of vectors that justify the theoretical problem raised? What is the distribution of the vectors characteristics? Do they really represent the search space around similarity and diversity? In summary, in order to actually quantify how general and applicable the proposal is, we need to understand which scenario characteristics benefit most from it.

Second, although the experiments seem to be sound and correct, I was very surprised that the authors employed just one baseline, and a classic one (MMR). It really constrains any assessment regarding its significance and generality on the base task. Further, some of the results presented in Table 2, although numerically the best, are not statistically significant considering the confidence intervals.

Finally, there are some typos in the paper, requiring some review and table 3 is too small, please increase the font size.

**Questions:**

- Please provide some characterization of the opportunity provided, considering actual datasets, for the proposed approach. In practice, we really face a potentially large scale NP-complete problem?

- Please consider more baselines that allow better evaluation of the effectiveness and generality of the proposal.

**Details Of Ethics Concerns:**

They use just publicly available datasets.

---

### Note · Authors · 2024-12-11

I have read and agree with the venue's withdrawal policy on behalf of myself and my co-authors.